# Protocol for a cluster randomised waitlist-controlled trial of a goal-based behaviour change intervention for employees in workplaces enrolled in health and wellbeing initiatives

**Lailah Alidu[1], Lena Al-Khudairy[2], Ila Bharatan[3], Paul Bird[4], Niyah Campbell[1], Graeme Currie[3], Karla Hemming[1], Kate Jolly[1], Laura Kudrna**  **[1]\*, Richard Lilford[1], James Martin[1], Laura Quinn[1], Kelly Ann Schmidtke[5], James Yates[1], On behalf of the Prevention Workplace Collaboration West Midlands[¶]**

1 Institute of Applied Health Research, University of Birmingham, Birmingham, England, 2 Warwick Medical School, Health Sciences, University of Warwick, Coventry, England, 3 Warwick Business School, Entrepreneurship & Innovation Group, University of Warwick, Coventry, England, 4 West Midlands Academic Health Science Network, West Midlands, England, 5 University of Health Science and Pharmacy, St Louis, MO, United States of America

¶ Authors listed in alphabetical order and are the complete list of members of the Prevention Workplace Collaboration West Midlands.
\* l.kudrna@bham.ac.uk

**Data Availability Statement:** No datasets were generated or analysed during the current study. All

## Abstract

Many workplaces offer health and wellbeing initiatives to their staff as recommended by international and national health organisations. Despite their potential, the influence of these initiatives on health behaviour appears limited and evaluations of their effectiveness are rare. In this research, we propose evaluating the effectiveness of an established behaviour change intervention in a new workplace context. The intervention, 'mental contrasting plus implementation intentions', supports staff in achieving their health and wellbeing goals by encouraging them to compare the future with the present and to develop a plan for overcoming anticipated obstacles. We conducted a systematic review that identified only three trials of this intervention in workplaces and all of them were conducted within healthcare organisations. Our research will be the first to evaluate the effectiveness of mental contrasting outside a solely healthcare context. We propose including staff from 60 organisations, 30 in the intervention and 30 in a waitlisted control group. The findings will contribute to a better understanding of how to empower and support staff to improve their health and wellbeing.

**Trial registration:** ISRCTN17828539.

relevant data from this study will be made available upon study completion.

**Funding:** This project is funded as part of the National Institute for Health and Care Research's (NIHR) National Priority Area Research Programme 2020-23 via the 'Prevention including Behavioural Risk Factors' Applied Research Collaboration (ARC) Consortium. The Consortium is led by the NIHR ARC North East and North Cumbria and ARC West Midlands. This project is hosted and led by the NIHR ARC West Midlands (NIHR200165) collaborating with ARC Northwest London and ARC North East and North Cumbria. The views expressed are those of the author(s) and not necessarily those of the NIHR or the Department of Health and Social Care. Use of the items adapted from the Diabetes Empowerment Scale means the project described was supported by Grant Number P30DK020572 (MDRC) from the National Institute of Diabetes and Digestive and Kidney Diseases. The funders had no role in study design, data collection and analysis, decision to publish, or preparation of the manuscript.

**Competing interests:** The authors have declared that no competing interests exist.

# Introduction

## Background and rationale

International and national organisations such as the World Health Organisation (WHO) and the United Kingdom Health Security Agency and Office for Health Improvement and Disparities, recommend utilising the workplace as a setting to prevent ill health and wellbeing [1, 2]. Despite enthusiasm for this approach, rigorous evaluations of workplace health and wellbeing initiatives (WHIs) are lacking, and they appear to have limited impact on changing employee health behaviours [3–8]. For example, a recent trial introduced a new, multi-component WHI that encouraged staff participation with financial incentives and found no change in behaviours related to physical activity [6]. The current research focusses on motivating employees within organisations already enrolled in WHIs to adopt behaviours that prevent ill health and wellbeing using well-established approaches to behaviour change called 'mental contrasting' and 'implementation intentions' [9–12].

The theory underlying mental contrasting is that mental contrasting encourages 'meta-cognition' with instructions that facilitate mental elaboration about how positive the future will be if a personal goal is attained and how the present reality gets in the way. By contrasting a positive future with a present reality, mental contrasting strengthens goal commitment, and it is more effective than thinking about a positive future (so-called 'indulging'). Mental contrasting is often combined with a technique called 'implementation intentions'; a self-regulatory strategy related to goal striving that encourages people to make 'if-then' plans related to their goals [13–16]. For example, if the goal is to eat less refined sugar, an 'if-then' plan might be, "If I feel like a sweet snack in the afternoon, then I will have an apple." In applications of mental contrasting and implementation intentions, participants are asked to select from multiple potential goals (called wishes) and strengthen their commitment to them using these strategies [17]. Further details on mental contrasting and implementation intentions, which include videos, printable handouts, a mobile phone app, and online exercises, can be found online (*woopmylife.org*).

Systematic reviews show that mental contrasting without and with implementation intentions is an effective health behaviour change intervention [11, 12]. A 2019 review [11] pooled ten studies with over 1,500 participants that examined mental contrasting alone. A random-effects model using measures from this sample at a four-week follow-up showed an overall combined effect size on health behaviour change (using adjusted Hedges g to measure effect size) of g = 0.28 (CI [0.13–0.43], p < .001, and an increased effect at up to three months, g = 0.38, CI [0.20–0.55], p < .001). A similar effect was observed when mental contrasting was combined with implementation intentions, g = 0.28 (CI [0.14–0.42], p < .001). These small-to-moderate effect sizes demonstrate meaningful change in real-world behaviours that are difficult to impact, including taking steps to stop smoking and eat healthier. Mental contrasting is thus a well-established, theory-driven, and evidence-based approach for empowering individual goal commitment and goal attainment.

Our research will be the first time, to our knowledge, that mental contrasting has been implemented alongside other WHIs outside of healthcare workplace settings. There are at least two recent systematic reviews on mental contrasting, including the one discussed above, but they do not distinguish workplaces from other settings [11, 12]. To establish our unique contribution and refine our approach, we conducted a systematic literature search of mental contrasting (without or with implementation intentions) interventions in the workplace (Prospero registration number CRD42022340063). See S1 File for the PRISMA flow diagram, databases, and search string [18].

The results of the search identified three studies testing the effects of mental contrasting in a workplace setting in randomised trials, and all the trials were in healthcare settings. In 2010, Oettingen and colleagues tested the effects of teaching mental contrasting versus indulging (fantasising about a positive future outcome only) with a sample of middle managers working in hospitals. The results showed that those taught to use mental contrasting enhanced their time management, decision-making, and project mastery [19]. In another study from 2010, Oettingen and colleagues looked at mental contrasting and help-giving behaviours in critical care paediatric nurses [20]. They examined whether mental contrasting produced an expectancy-dependent (that is, feasibility-related) commitment to giving help to others. They found that mental contrasting induced commitment to help-giving but only when critical care nurses perceived it as feasible. In 2018, Gollwitzer and colleagues evaluated an online mental contrasting and implementation intentions intervention aimed at stress reduction in hospital nursing staff [21]. The results showed that staff in the intervention group reported reductions in stress and enhanced work engagement relative to a no-treatment control group and another a modified intervention group who made more specific plans.

We can thus assert that our study is the first to assess whether mental contrasting is effective in a sample including non-healthcare staff. This is important because workplaces are a key setting for public health interventions as they afford access to large groups of people who may not otherwise engage with healthcare organisations [1, 2]. We cannot assume the results will generalise to other sectors because healthcare workers have more knowledge of health and wellbeing and may react differently to the intervention [22–24]. Our study complements other research on goal setting for health and wellbeing that embeds goal setting alongside other approaches but does not use mental contrasting and implementation intentions, such as in-depth counselling [25] or motivational interviewing [26] (see [27] for a review including studies of goal setting in promoting physical activity in the workplace).

To add to the existing evidence on the effectiveness of mental contrasting in workplaces, we will conduct a waitlisted cluster randomised controlled trial evaluated with a mixed methods approach. The purpose of the waitlisted approach is to highlight the social appeal taking part, as individuals and organisations may not wish to take part if there is a risk of being in the control group and not receiving any intervention. Our outcomes are self-reported, and the intervention period lasts only several weeks. If positive effects are found, a more rigorous trial could consider including objective measures of health and extending the timeframe. This trial will be embedded in an ongoing WHI run by a local government team. It is part of a larger programme of research looking at how the wider context and mechanisms shape outcomes within WHIs and how employers can support a culture of staff self-care behaviours.

## Objectives

Our objective is to establish the effectiveness of a mental contrasting and implementation intentions intervention that is delivered through workplaces on employees' self-rated progress towards achieving their health and wellbeing goals.

We will answer three research questions:

1. Does the intervention result in participants perceiving they have made progress towards achieving their health and wellbeing goals?

2. Does the intervention change employees' perceptions of their health, health behaviour, and wellbeing?

3. What barriers, enablers, and mechanisms do employees perceive as related to the effectiveness of the intervention?

For research questions (1) and (2), we will also conduct sub-group analyses to establish whether the intervention is more effective according to demographic and workplace characteristics that were suggested by the Patient and Public Involvement (PPI) Group and members of the research team, including: gender, household income, ethnicity, disability, self-rated health, occupational role, working from home, health risks at work, duration in job role, and organisational size.

## Methods: Participants, interventions, and outcomes

This protocol has been prepared according to the SPIRIT checklist for the content of protocols for randomised controlled trials [28, 29]. See S2 File.

### Trial design

An embedded mixed methods cluster randomised waitlist-controlled trial with workplaces as the cluster. The embedded approach reflects that the qualitative data will be used to enhance and explain the quantitative results [30].

### Study setting

Workplaces in the Coventry local authority region of West Midlands, England, United Kingdom, who are signed up to a local government WHI. The WHI provides information and advice on supporting staff health and wellbeing and a system of accreditation for organisations who demonstrate progress towards achieving health and wellbeing criteria [31].

### Eligibility criteria

Workplace eligibility criteria (cluster-level):

1. Signed up to participate in the Coventry WHI

2. Willing and able to allow at least ten employees to participate in data collection activities

   Employee eligibility criteria (individual-level):

1. Aged 16 years of age or older

2. Willing to provide written consent

### Study design

The embedded mixed methods cluster randomised waitlist-controlled trial design will be realised with all participants being invited to attend two group sessions. The sessions will include both the data collection and intervention activities. The schedule of activities for these sessions is in Table 1. In Session 1 for the intervention group, they will receive a welcome, fill out the quantitative survey, have a general group discussion about health and wellbeing at work, and then receive the intervention (see below). Session 1 for the control group is identical except they do not receive the intervention. In Session 2, all participants will again receive a welcome and conduct a quantitative survey. The intervention group will then take part in a qualitative focus group about mechanisms, barriers, and enablers, whereas the control group will receive the intervention. Note that all surveys are carried out prior to discussion in both sessions, meaning endpoint measures will be taken from the control group before they receive their intervention.

**Table 1. Schedule of activities for group sessions about health and wellbeing, which contain data collection and intervention activities.**

| Session 1: Intervention group | Session 1: Control group |
|---|---|
| a. Welcome | a. Welcome |
| b. Quantitative survey (baseline assessment) | b. Quantitative survey (baseline assessment) |
| c. Group discussion on health and wellbeing at work | c. Group discussion on health and wellbeing at work |
| d. Intervention | |
| **Session 2** *(4+wks after Session 1)***: Intervention group** | **Session 2** *(4+wks after Session 1)***: Control group** |
| a. Welcome | a. Welcome |
| b. Quantitative survey (endpoint assessment) | b. Quantitative survey (endpoint assessment) |
| c. Group discussion on mechanisms, barriers, enablers | c. Intervention |

Note that the timing of activities with sessions corresponds with letters a-d (where a is first, b is second, etc.)

## Interventions

**Description of intervention.** The intervention is mental contrasting plus implementation intentions. It will be delivered to employees from the same organisation in a group setting during working hours. The instructions that will be used to guide the delivery of the intervention and accompanying participant handout is enclosed in S3 File, which is derived from existing studies on mental contrasting and implementation intentions [32–34]. We enclose a more detailed description of the intervention in S4 File along with the completed TIDieR checklist [35]. In brief, all participants will be invited to attend a group session about health and wellbeing with other employees that lasts around one hour. All participants will be asked to think of something that they would like to do for their health and wellbeing (a 'wish') in the next four weeks that is important, achievable, and challenging. Part of the group session will contain the intervention activity (see Table 1). During the intervention activity, participants will be encouraged to use mental contrasting and implementation intention steps to reinforce the wish, turning it into a committed goal. Participants will also be provided with a handout that shows the steps and has space to practice using them.

**Intervention delivery.** The group sessions will take place in person or virtually, depending upon the preferences of the workplace and any restrictions on contact due to the ongoing COVID-19 pandemic. The intervention will be delivered to groups of staff by a researcher (MSc-level education or higher) familiar with mental contrasting and implementation intentions, and, if available, supported by a workplace champion. Organisations intending to include their health and wellbeing champion will require them to attend a session organised by the research team prior to intervention delivery. The approach to the delivery method was co-created with the PPI group through an idea generation and ranking exercise.

## Quantitative measures

The primary quantitative outcome measure is self-reported progress towards goal attainment. At endpoint, participants will be reminded that they had been asked to identify something that they wished to do for their health and wellbeing. Then, they will be asked about their self-reported progress: "So far, how much progress would you say that you have made towards what you wished to do for your health and wellbeing?" with responses ranging from no progress to a lot of progress (1–7). This item was adapted by the PPI group from prior research [36] and addresses objective one.

As a secondary outcome, which was suggested by the PPI group, participants will be asked a related question about their behavioural changes, "And how much progress in *changing your*

*behaviour* would you say you have made towards what you wished to do for your health and wellbeing?" with responses ranging from no progress to a lot of progress (1–7). This question partly addresses objective two.

Additional secondary quantitative outcomes will be assessed to broadly explore participant perceptions of their heath, health behaviours, and wellbeing more fully. Participants will be asked to self-rate their health using an established measure [37]. Measures of empowerment to change their health and wellbeing (attitudes, knowledge, and behaviour) will be adapted from prior empowerment scales [38–40]. Measures of psychological wellbeing will include items about job satisfaction, hedonic wellbeing (e.g. pleasure, pain, happy, anxious, joy) and eudemonic experiences (e.g. meaning, futility, worthwhileness, pointlessness), and mental wellbeing on the short form of the Warwick Edinburgh mental wellbeing scale [41–45]. These wellbeing items also address objective two.

We will also include items on demographic and workplace characteristics: self-reported sex and gender, household income, ethnicity, disability, health, occupational role, home working, health risks at work, duration in job role, and organisational size. The purpose of these items is to ensure that we capture the diversity of our participants and to assess potential inequalities in the effectiveness of the intervention, as suggested by the research team and our PPI group.

Our quantitative measures were informed by a meeting with researchers, the local government team, and PPI representatives using guidance from the What Works Centre for Wellbeing on modelling the relationships between individual and community wellbeing. This guidance focusses attention on different levels of individual and workplace communities, including unintended outcomes and sub-group differences [46]. We also drew on a logic model from prior research with the local government team [47], modified and shown as a causal chain in a diagram S5 File. In line with theory on causal reasoning, we show our quantitative observations across a causal chain going from the intervention to employees' perceived empowerment to change their health and wellbeing, employee perceptions of progress towards their health and wellbeing goals, employee behaviour change, and subjective wellbeing [48]. Note that employee behaviour change and wellbeing may occur as a result of largely automatic and unconscious attentional processes, bypassing conscious perception, which the diagram reflects [49, 50].

## Qualitative measures

Qualitatively, we will use focus groups to identify and discuss analytic themes about barriers and enablers to the effectiveness of the intervention, as well as themes about implementation processes to understand if the intervention becomes adopted as part of normal practice by groups of employees. The qualitative research addresses objective three. Our semi-structured topic guide (S6 File) is informed by concepts from normalisation process theory, which describes how interventions are adopted, implemented and sustained [51–53]. Several questions about mechanisms are adapted from questions from the normalisation process toolkit [53]. We explore aspects of the intervention implementation, such as the difference between what might be currently available, the purpose of the intervention and how it may influence their health (coherence), how to improve and sustain the intervention (cognitive participation), incorporating the intervention in work and daily life and support by organisation (collective action), and explore if the intervention is perceived as worthwhile (reflexive monitoring). All groups will also answer general question about health and wellbeing at work related to the programme offered by the local government team.

## Participant timeline

The sessions described in detail in Table 1 are shown according to the SPIRIT schedule of enrolment, intervention, and assessment in Fig 1. A brief summary timeline is provided in S7 File. The intervention period lasts a minimum of four weeks, consistent with prior research on mental contrasting [11, 54, 55]. After consenting to take part, half of the organisations will be randomised to receive the goal-setting intervention (Group 1). The other half (Group 2 - wait-list control group) will receive no intervention until after the first group completes the goal-setting intervention.

| 2022/23 TIMEPOINT | Enrolment | Allocation | Post-allocation | | Close-out |
|---|---|---|---|---|---|
| | *-t₁* | **0** | *t₁* | *t₂* | |
| | *May '22-Feb '23* | *Sep '22-Feb '23* | *Oct '22-Feb '23* | *Nov '22-Mar '23* | *tₓ* |
| **ENROLMENT:** | | | | | |
| **Eligibility screen** | X | | | | |
| **Informed consent** | X | | | | |
| **Invitation to group sessions** | X | | | | |
| **Allocation** | | X | | | |
| **INTERVENTIONS:** | | | | | |
| *[Intervention group]* | | | X | | |
| **ASSESSMENTS:** | | | | | |
| *Workplace characteristics: industry and size* | X | | | | |
| *Secondary outcomes: psychological wellbeing, demographic and workplace characteristics* | | | X | X | |
| *Primary outcome: self-reported general progress towards goal* | | | | X | |
| *Secondary outcome: self-reported behavioural progress towards goal* | | | | X | |

**Fig 1. SPIRIT schedule of enrolment, intervention, and assessment.**

## Sample size

A sample size of 60 workplaces is planned to participate in the trial based on what the local government team have stated as feasible, they will be allocated equally between the intervention and control group. Data from a previous cluster randomised trial [56] that included 100 workplaces was used for the estimating average cluster sizes and the coefficient of variation of cluster sizes. Data from a previous study [36] with a similar outcome was used to calculate the standard deviation of the outcome. No data or published trial results were available from which to estimate or inform intra-cluster correlations coefficients (ICCs), so in place as recommended we have used a range of plausible values [57].

Our sample size calculation is based on the main continuous outcome (how much progress towards goal achievement, measured on a seven-point scale), a parallel cross-sectional design, an exchangeable correlation structure, an average cluster size of 10, varying cluster size (coefficient of variation: 0.30), likely estimates of the ICCs (0.01, 0.05, 0.10) and a standard deviation of 1.55 in the progress towards achievement scale implemented using design inflation factors for the cluster randomised nature of the design [58]. Over 80% power is available across all scenarios to detect a half-point increase in the outcome between the intervention and control group (see Table 2). A difference of a half-point was considered clinically important because it can be difficult to change health behaviours even when people intend to make a change [59].

## Recruitment

The expected recruitment dates are October 2022 through February of 2023. Participants will be recruited through our collaboration with the local government team. There are approximately 60 organisations currently participating in the local government WHI in Coventry and these workplaces will be approached by their regular point of contact, the Business Development Advisor, Employment and Wellbeing Service, Coventry City Council. The organisations will be invited to participate in the trial. The key contacts in organisations will be asked to invite volunteer employees to participate until there are at least 10 and no more than 30 employees per organisation (in both Groups 1 and 2). All available and willing participants will be added until 30 participants are reached. To encourage a diverse sample from each organisation, the key contacts will be asked to invite employees who represent diverse occupations throughout the organisation (such as those who work remotely and in person, on and off site, in offices and on shop floors), but to avoid inviting any senior level managers or board members (to avoid the perception that employees feel as if they need to take part). Key contacts within organisations will be provided with study information and asked to sign an organisational commitment to the research, which is shown in S8 File, and provide brief details about their organisation's size and sector.

**Table 2. Power achievable under anticipated available sample size over plausible scenarios.**

| Mean difference | ICC = 0.01 | ICC = 0.05 | ICC = 0.10 |
|---|---|---|---|
| 1 | 1.00 | 1.00 | 1.00 |
| 0.5 | 0.96 | 0.90 | 0.80 |
| 0.25 | 0.47 | 0.37 | 0.29 |

## Methods: Assignment of interventions (for controlled trials)

### Randomisation, allocation, sequence generation

All workplaces will be assigned an anonymous ID. An independent statistician will generate an allocation sequence using the random number generator in Stata v16.1 with assignments to the intervention or control group stratified by company size. Random block sizes of two and four will be used to maintain both balance within stratum and prevent lack of allocation concealment. A company will be considered small if there are less than 50 employees, medium is 51 to 250 employees and large is more than 250 employees [60]. The stratification variable was chosen based on a previous study [47]. The independent statistician will retain this list and it will be kept concealed from the study team. After baseline data collection has occurred in each workplace, its assignment will be revealed based on the next assignment in the sequence for its particular stratum.

### Blinding (masking)

Researchers collecting quantitative data will be blind to trial group allocation at baseline. After data collection and before intervention delivery, researchers will open an envelope to determine if the workplace is in the intervention group. Participants will not be told if they are in the intervention group. Some quantitative data collection will occur electronically, and this data collection mode does not involve researchers who are aware of the trial group.

## Methods: Data collection, management, and analysis

### Data collection methods

Quantitative data collection will occur immediately before and at least four weeks after the intervention (see Table 1 and S7 File). Data collection will be conducted virtually (using Qualtrics e-surveys), face-to-face (via pen/pencil and paper), or over the phone (with researchers inputting participants' responses into Qualtrics e-surveys), depending on the preferences of the workplaces and their staff. Qualitative data about the intervention will be collected in Session 2 (see Table 1 and S7 File), after quantitative endpoint data collection and at least four weeks after the intervention. We anticipate collecting qualitative data via virtual focus groups (unless another mode is preferred), and quantitative data via online e-surveys at the beginning of the focus groups. If participants are not able to attend their second session, they may be emailed a link to the e-survey or called to complete it over the phone. The e-survey questionnaire is in S9 File, and the qualitative focus group discussion guide is in S6 File. Sessions, including the data collection tools, will be piloted with an organisation not in the trial.

### Data management

Qualitative recordings will initially be stored on the Cloud of [anonymous for peer review] associated with Zoom or Microsoft Teams. Recordings may also be made in password-protected handheld recording devices. All interview links will be password-protected. Data from quantitative online interviews will be exported from Qualtrics survey platforms and stored on secure [anonymous for peer review] servers. Quality checks will be performed on the quantitative data, including checking for the proportion of missing data and plausible values.

### Analysis of quantitative data (statistical methods)

All analyses will be intention to treat, meaning that workplaces (clusters) will be analysed according to their randomisation allocation. Baseline characteristics will be summarised by

intervention and control group. Means and standard deviations or medians and interquartile ranges will be used to summarise numeric characteristics, as appropriate. Frequencies and percentages will be used to summarise categorical characteristics. A flowchart will be created to show the flow of the workplaces and employees throughout the trial.

The primary comparison will be between clusters randomised to the mental contrasting intervention versus clusters randomised to the control. For the primary outcome, whether an employee has made any progress towards their wish (measured on seven-point scale), a Bayesian mixed effects linear regression model will be fitted to calculate the mean difference between the intervention and control group. Any participants who did not remember their wish will be coded into the lowest point category. Assumptions for linear regression will be checked and data transformations will be considered if necessary. The model will include an intervention indicator, covariate used in the randomisation (cluster size) and cluster as a random effects term. A Bayesian approach will be used for the incorporation of prior uncertainty, assigning "uninformative vague" prior distributions to the model parameters with a mean of 0 and a standard deviation of 10,000. For the primary analysis, observations with missing outcome data will be excluded.

Additionally, secondary analysis will be completed fitting a fully adjusted model for the primary outcome which will adjust for the following covariates: gender; household income; ethnicity; disability; self-rated health; occupational role (professional versus non-professional [61], see questionnaire in S9 File); home working; health risks; duration in job role and organisation size. Multiple imputation will be completed if there is more than five percent missing covariate data [62]. Multiple imputation will allow for the clustered nature of the data and the number of imputations will be dependent on the percentage of missing data.

All other secondary outcomes are on a similar or same scale to the primary outcome and therefore will take the same format as the analysis above. For secondary outcomes such as self-reported health, models will have an indicator for baseline or endpoint measures.

Subgroup analysis will be completed for the primary outcome by workplace and demographic characteristics, using interactions tests. Demographic and workforce characteristics will include gender (male/female/other), household income (less than 10K, 10-20K, 20-30K, 30-40K, 40-50K, 50-60K, 60-70K, 70K+), ethnicity (White; Mixed or Multiple Ethnic Groups; Asian or Asian British; Black, Black British, Caribbean or African, Other, Prefer not to state), disability (yes - specify, no), self-rated health (poor, fair, good, very good, excellent), occupational role, home worker, health risks at work ("I feel physically safe at work?" completely agree, mostly agree, slightly agree, slightly disagree, mostly disagree, completely disagree), duration in job role (less than six months, six months to a year, more than a year) and organisation size (small, medium, large).

All estimates will be reported with 95% credible intervals and posterior probabilities for any benefit will be reported (mean difference>0). All analyses will be completed in Stata v16.1.

### Analysis of qualitative data

Qualitatively, focus groups will be audio recorded, transcribed, and coded, and reflexive thematic analysis will be used to conceptualise analytic themes about barriers, enablers, and mechanisms surrounding the use of WOOP [63–65]. We will also analyse fieldnotes from focus groups. To inform the development of themes, will draw on the behaviour change wheel [24] to deductively shape our understanding of the data on barriers and enablers and on normalisation process theory to shape themes from the data on mechanisms [51]. Our coding approach will involve members of the research team who both did and did not collect the data, mixing informed and independent approaches, taking a collaborative and reflexive approach

to identify patterns of shared meaning across the dataset [65]. Our results will be reported using the consolidated criteria for reporting qualitative research [66].

## Methods: Monitoring

### Data monitoring

A statistician will conduct a quality check on the data for completeness and plausibility midway through each of the baseline and endpoint data collection periods.

### Harm

Participants are informed that there are minimal risks involved with participation in the research and that they can contact the lead investigator. If participants are experiencing mistreatment at work, researchers will signpost them to their Human Resources department or [anonymous for peer review].

### Auditing

If participants have any concerns or complaints about the research and trial process, they are told that they can contact [anonymous for peer review].

## Ethics and dissemination

### Research ethics approval

Ethical approval was obtained from University of Birmingham Science, Technology, Engineering and Mathematics Ethical Review Committee - ERN_21–0744.

### Protocol amendments

All amendments to the study will be sent to the Ethics Review Committee for approval before implementation and the trial registration will be updated.

### Consent or assent

Consent will be obtained from the employees who will participate in the trial by researchers. Consent will be in a written or electronic document signed by participating employees (see S10 File for the Participation Information Sheet and Consent Form).

### Confidentiality

Participants' data will be treated as confidential (i.e., they will not be identified in any outputs from the study, and their identity will not be disclosed to any third party). Data will be pseudo-anonymised. Password-protected files and folders stored on [anonymous for peer review] servers will contain the consent forms, participant names, and an ID code to link to participants' data. Qualitative audio/video files and quantitative datasets will be password-protected and stored with the ID code.

### Access to data

The anonymised participant-level quantitative data and analysis scripts will be available upon request to [anonymous for peer review] upon publication in a peer-reviewed journal.

### Ancillary and post-trial care

A debriefing information sheet will be sent to workplaces that they can use to communicate with staff about the trial (see S11 File).

### Dissemination policy

Our dissemination plan will be co-created with the PPI group to identify key audiences and appropriate modes of communication. Workplaces will receive a copy of the final publication. Authorship eligibility will be determined according to the International Committee of Medical Editors (ICMJE) [67].

## Strengths and limitations

Some of the strengths of this research are its randomised design, which allows causal inference, and its mixed methods approach and allows the investigation of implementation and mechanisms. There are several limitations to this research. One of these is generalisability, as it is may be that employees in organisations who are already signed up to health and wellbeing initiatives have better health and wellbeing. There could also be interactions between the intervention and other ongoing initiatives within the workplaces. However, it is important that individual-level behaviour change initiatives are provided in a supportive context alongside other initiatives or they may fail in isolation and when they do not address social and structural determinants of health [68]. It could also be that collecting baseline data immediately prior to the intervention limits any effect of the intervention in other contexts where baseline data is not collected beforehand.

A further limitation is that employees volunteered to take part, which may introduce selection bias. Although a mitigation step is planned by asking workplaces to invite staff across diverse occupations in workplaces, purposeful selection in this way means that the results may not generalise to those who are not selected. Blinding is not fully possible because participants will know they are part of a research study, although they will not be explicitly told if they are in the intervention or control group. The dose of the intervention in one group session might be too weak and repeated practice or reminders over the study period could strengthen its effects. Multiple doses for various wishes could be required to achieve results. Finally, all outcomes are self-reported. This is a limitation because no conclusions can be drawn about objective effects on health or health behaviour and there could be response bias. If positive effects are found for measures of self-report, future research could examine the effects on observational measures like absenteeism or productivity, if any changes are sustained over the longer term, or if changes have clinically significant implications for health and wellbeing.

## Supporting information

**S1 File. Systematic literature search PRISMA flow diagram, databases, and search string.** (DOCX)

**S2 File. Completed SPIRIT checklist.** (DOCX)

**S3 File.** a. Session 1 - Part 1 –all. b. Session 1 - Part 2 - intervention only. c. Session 1 - Part 2 - control only. d. Session 2 - intervention only. e. Session 2 - control only. f. Handouts. (ZIP)

**S4 File. TIDieR checklist and further intervention information.** (DOCX)

**S5 File. Causal chain from intervention to outcomes.**
(DOCX)

**S6 File. Semi-structured focus group discussion guide.**
(DOCX)

**S7 File. Brief summary timeline.**
(DOCX)

**S8 File. Organisational information sheet and commitment.**
(DOCX)

**S9 File. Quantitative survey questionnaire.**
(DOCX)

**S10 File. Participant information sheet and consent.**
(DOCX)

**S11 File. Debriefing information for workplaces.**
(DOCX)

**S12 File.**
(ZIP)

## Acknowledgments

The Prevention Workplace Collaboration West Midlands thanks contributors from our colleagues in the National Institute for Health and Care Research (NIHR) Applied Research Collaboration (ARC) Northwest London and ARC North East and North Cumbria, especially Austen El-Osta, Immy Webber, Aos Alaa, Mackenzie Fong, Charlotte Rothwell, Claire O'Malley, Pat Watson, Frances Hillier-Brown, and Amelia A Lake. We are also grateful for support from our policy partners at Coventry City Council, especially Jenny Duggan and Laura Howard. Thank you to our group of public contributors, including Sarah Markham, Danny Gregory, Samantha Russell, Graham Brown, and Anne Phillips.

## Author Contributions

**Conceptualization:** Lailah Alidu, Lena Al-Khudairy, Ila Bharatan, Graeme Currie, Kate Jolly, Laura Kudrna, Richard Lilford, Kelly Ann Schmidtke, James Yates.

**Formal analysis:** Karla Hemming.

**Funding acquisition:** Lena Al-Khudairy, Ila Bharatan, Niyah Campbell, Graeme Currie, Karla Hemming, Kate Jolly, Laura Kudrna, Richard Lilford, Laura Quinn, Kelly Ann Schmidtke.

**Investigation:** Lailah Alidu, Niyah Campbell, Laura Kudrna, Richard Lilford, James Yates.

**Methodology:** Karla Hemming.

**Project administration:** Niyah Campbell, Kate Jolly, Laura Kudrna, Richard Lilford, Kelly Ann Schmidtke.

**Resources:** Lailah Alidu, Niyah Campbell, Laura Kudrna, Richard Lilford, Laura Quinn, James Yates.

**Software:** Karla Hemming, Laura Quinn.

**Supervision:** Paul Bird, Kate Jolly, Laura Kudrna, Richard Lilford, Kelly Ann Schmidtke.

**Writing – original draft:** Lailah Alidu, Laura Kudrna, Richard Lilford, Laura Quinn, James Yates.

**Writing – review & editing:** Lailah Alidu, Lena Al-Khudairy, Ila Bharatan, Paul Bird, Niyah Campbell, Graeme Currie, Karla Hemming, Kate Jolly, Laura Kudrna, Richard Lilford, James Martin, Laura Quinn, Kelly Ann Schmidtke, James Yates.

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
