## [Decision Letter · Decision Letter 0]

16 Aug 2022

PONE-D-22-11119

A cluster randomised waitlist-controlled trial of a goal-setting behaviour change intervention for employees in workplaces enrolled in workplace health and wellbeing initiatives

PLOS ONE

Dear Dr. Kudrna,

Thank you for submitting your manuscript to PLOS ONE. After careful consideration, we feel that it has merit but does not fully meet PLOS ONE’s publication criteria as it currently stands. Therefore, we invite you to submit a revised version of the manuscript that addresses the points raised during the review process.

The manuscript has been evaluated by two reviewers, and their comments are available below. Please address all of the reviewers' comments in your revisions and Response to Reviewers document. If there are instances where the reviewers make opposing recommendations, please discuss this in the Response to Reviewers and provide a clear explanation of how you have responded.

The submission does not currently include a description of how you plan to validate the new survey tool developed for the quantitative data collection before using the tool for the full study. Please discuss this in your revision.

In addition to responding to the reviewers, please address the following points which pertain to the journal's requirements:

When submitting to PLOS journals, please ensure that the full author list is entered into the Editorial Manager form at the time of submission. This is important in enabling the journal to complete our checks and ensuring that co-authors receive the relevant manuscript notifications. Please update the Editorial Manager submission details before you upload your revise manuscripts, and indicate clearly in your response letter whether the final article, if accepted, should list the consortium name  or the individual author names in the published author list. This article was submitted in May, 2022, and the submission indicates that the study was submitted for ethical review and that expected recruitment dates were June through August, 2022. Please clarify whether you have obtained ethics approval, and either provide updated recruitment dates or indicate in your cover letter if you have already begun the recruitment process. Please note that according to PLOS ONE's requirements (https://journals.plos.org/plosone/s/what-we-publish#loc-registered-reports), Registered Report Protocols must be published before the study (including recruitment) begins. If you have already begun recruitment then contact the journal (plosone@plos.org) to discuss whether it would be suitable to change the article type to a Study Protocol.Please clarify whether the quantitative survey tool includes any items that are copied directly from or that are adapted from previously published tools. Note that if sources for these items were not published under CC BY, copyright restrictions may apply such that you would need to obtain license/permissions to include those questions in your Supporting Information file. Please email plosone@plos.org to discuss this matter if this applies in the case of your article.Thank you for noting that images in Appendix 3 are from www.pexels.com. PLOS content is published under a CC BY license, and we need to ensure that this can be applied to all materials included with your submission. Please clarify the following: The Terms of Service indicates that contents from Pexels is either subject to a CC0 License or to the Pexels License described in section 5.5 of the Terms of Service. Which images used in Appendix 3 are subject to CC0 vs. Pexels Licenses? The Pexels License grants permission to distribute the content but does not specifically mention publication of the images. Please obtain from the company a clarification as to whether the License includes permission to publish the images under a CC BY license. Include the company's response in a Supporting Information file with your revision.The Pexels Terms of Service (https://www.pexels.com/terms-of-service/) says, "5.7. Be aware that, depending on your intended use of the Content, you may need the permission or consent of a third party (e.g. owner of a brand, identifiable person or author/rights holder of copyrightable work depicted in the Content)." For the two images in Appendix 3 that obtain identifiable people, are permissions needed to include the image in a CC BY publication, according to the license and/or other information available on pexels.com for those images?

A marked-up copy of your manuscript that highlights changes made to the original version. You should upload this as a separate file labeled 'Revised Manuscript with Track Changes'.An unmarked version of your revised paper without tracked changes. You should upload this as a separate file labeled 'Manuscript'.

We look forward to receiving your revised manuscript.

Kind regards,

Renee Hoch, Ph.D.

Managing Editor, Publication Ethics

PLOS ONE

Journal Requirements:

2. Please confirm that you have received IRB/REC approval, and update both the ethics statement in the submission form and the methods section in your manuscript with the name of the ethics committee that approved your study.

3. Please enter into the EM submission form all your data, or data for the The Prevent Workplace West Midlands Collaboration if this group should be listed on the article instead of individual authors. 

Reviewers' comments:

Reviewer's Responses to Questions

**Comments to the Author**

1. Does the manuscript provide a valid rationale for the proposed study, with clearly identified and justified research questions?

Reviewer #1: Partly

Reviewer #2: Yes

2. Is the protocol technically sound and planned in a manner that will lead to a meaningful outcome and allow testing the stated hypotheses?

Reviewer #1: Partly

Reviewer #2: Partly

3. Is the methodology feasible and described in sufficient detail to allow the work to be replicable?

Reviewer #1: Yes

Reviewer #2: Yes

4. Have the authors described where all data underlying the findings will be made available when the study is complete?

Reviewer #1: Yes

Reviewer #2: Yes

5. Is the manuscript presented in an intelligible fashion and written in standard English?

Reviewer #1: Yes

Reviewer #2: Yes

6. Review Comments to the Author

You may also provide optional suggestions and comments to authors that they might find helpful in planning their study.

Reviewer #1: This manuscript describes a proposed study design and protocol aiming to evaluate a mental contrasting intervention for achievement of health and well-being goals in the workplace.

In contrast to the stated rationale of showing an effect on health behaviour (p. 3-4), the objectives of the study (p.5) are all concerned with the participants’ perception of health and health-related behaviour. Indeed, the main outcomes are participants’ self-reported evaluation of their progress with respect to the goals they set at baseline. Several factors may be expected to influence these perceptions, including the expectations which the participants associate with the intervention, to which they are not blinded. Therefore, the results will not allow direct conclusions concerning the actual, objective effects on health and health behaviour, which will not be determined.

While these limitations apply to many studies especially of psychological interventions, the very subjective nature of the outcomes and the fact that participants are fully aware of the aims of the study and of their allocated intervention make them especially acute in this proposed study. They might be mitigated by veiling the study aim for the participants prior to the intervention, as seems to have been done by Oettingen, Mayer and Brinkmann (Ref. 18), and/or by asking the participants to assess their current health status and behaviour rather than related to baseline. The use of a concomitant ‘placebo’ intervention in the wait-list group might help to reduce disparity in expectations in the two groups. Failing this, these limitations should be described more fully and the planned procedure and questionnaire should be justified in detail.

It is important that the choices of self-set goal at baseline are broadly comparable between the intervention and waitlist groups. If possible, participants should remain blind to allocation until these goals habve been set. At data analysis, choices should be presented descriptively (e.g. by building categories) and compared between groups.

Minor comments:

1. Please explain why a Bayesian (linear mixed) model is planned, since there does not seem to be a concretely relevant prior for the effect.

2. Multiple imputation (p.15) is proposed for dealing with missing covariates in a secondary analysis. It would be good to give a reference for this technique.

3. A statistician is supposed to check data for accuracy (p.16) – is this possible, or do you mean plausibility?

Reviewer #2: Review of Registered Report Protocol PONE-D-22-11119 submitted to PLOS ONE entitled: “A cluster randomised wait-list controlled trial of a goal-setting behavior change intervention for employees in workplaces enrolled in workplace health and well-being initiatives”.

The present protocol describes an intervention study geared at supporting behavior change in the workplace context. The intervention employs a behavior change strategy called WOOP (Wish, Outcome, Obstacle, Plan) which in the scientific literature stands for mental contrasting (MC or WOO) by itself or combined with implementation intentions (MCII or WOOP). Mental contrasting with or without implementation intentions is well tested in the health domain but hardly in the work domain. In fact, the proposal argues that the innovation of the proposed study is that only three studies focus on mental contrasting or WOOP in the workplace and has never been systematically examined in a broad sample of workers during their everyday lives. Impressively, the authors want to address 60 organizations with 30 organizations in the intervention group and 30 organizations in the waitlisted control group. The wishes are supposed to tackle desired futures from the health and well-being domains.

The protocol includes a host of details that is relevant for administrative purposes, is well written, and gives an overview about what the authors are planning to do. The research question asking for the acceptance and effects of WOOP in the workplace is very important, and because the protocol aims at a relatively large sample of organizations and participants, the study is well worth to be conducted. The materials are in the most part well described.

Unfortunately, I am a bit unclear about my role as a reviewer. Judging from the timetables, the study has already started. The question then is, how much can the authors still change in the protocols? I think the study should be done, especially after all the effort of obtaining the contacts to the organizations. However, they are still a host of details which should be improved when it comes to design and instructions of the intervention. In the following, I provide examples on how the materials could be refined according to the theoretical and empirical principles of WOOP. I hope refinement is still possible before the beginning of the study.

I might not fully understand in what order and in what time frame the instructions are given. Do the authors assume that they can have the baseline filled out and the WOOP exercise taught in 45 minutes to an hour? Have the authors piloted that time frame? I'm also unclear how many participants should be in one group, how the researchers are trained to convey WOOP, how often WOOP is practiced in the first instruction session, and how the materials (participants take them home?) are explained. Plus, what are the fidelity criteria, etc.

It would be important that the interventionists strictly adhere to the instructions and that the instructions are better standardized. WOOP is a sensitive animal and small changes in wording might have big effects, as the experimental data show. In that respect, as nice as it is that the authors want to give freedom in delivery to the interventionists and want to leave the setting unspecified (Zoom, in person), there is the danger of diluting the WOOP effects. Experimental control should be guaranteed, at least as much as possible. Certainly, we need a balance between the autonomy of the interventionist and the sensitivity of the WOOP exercise, but it would be helpful to twist the WOOP instructions and examples (especially at the beginning) more in the direction of being in line with the WOOP principles plus towards being standardized. To find a compromise between standardization and autonomy of the interventionist the authors may consult the woopmylife.org website. Here one can also speak to the researchers who generated the strategy.

Let me specify some points to change in the instructions (without that I can be exhaustive here):

First the authors talk about a goal setting intervention. WOOP, however, starts with the Wish, not with the goal, and the research on mental contrasting makes a strong point that people generate wishes and concerns about desired futures rather than goals. In fact, only after mental contrasting wishes might turn into committal goals. It is critical that participants understand that they come up with a wish, that they are searching for. For finding their wish, participants need to be calm and slow, need to understand that their wish should be “dear to their heart” and truly important. By simply naming a goal participants will not achieve finding this wish (as we have so many goals on the top of our heads). The first step in WOOP, then, is foraging for a wish. Please consult the woopmylife.org website for relevant instructions. The authors can download the WOOP kit and for the tone and slowness also watch the instructional videos. In the instructions of the present protocol, the introduction of the goal at the beginning and then making the first WOOP regarding that goal might mislead people.

In the same vein, it would be helpful if the authors relinquish the term goal setting. The term goal often implies being overladen with pressure and demands from the outside world. Therefore, if the study still can be changed, it would be helpful to let go from the term goal setting when introducing the instructions. Also, to not confuse people, the term WOOP instead of MCII might be enough. Two terms are an unnecessary load.

When introducing WOOP, the examples do not comply with the WOOP principles. They should be referring to wish, outcome, obstacle, and behavior to overcome the obstacle. These four steps should be built on each other. And they should be short and concise (later the instructions talk about 3 to 6 words). Again, please check the woomylife.org website.

Later the instructions are more in line with the WOOP principles (p. 56 left side). The authors may want to adjust the examples and instructions, make them consistent across the study, and consistent with what has been depicted on the website.

But even in the detailed WOOP instructions, the authors might check again: For example, on p. 51 in the instructions the step to find and specify the behavior to overcome the obstacle is missing. This is a crucial step and should be integrated. Or, in the booklet, it is unclear what people need to do, how often, when, etc. Why does the wish have so many lines? P. 56, too many if-then plans might hurt.

The preparation for the WOOP exercise is missing. To do a WOOP exercise, people need to be slow and calm, and the instructions before the exercise need to explicitly say that. Otherwise, participants will not generate a meaningful wish and will not be able to come up with the best outcome, and internal critical obstacle, nor will they find a behavior to overcome the obstacle. Importantly, they will not be able to vividly imagine the outcome and the obstacle either.

It is crucial that people understand that the obstacle needs to be an inner obstacle, an obstacle such as an emotion, belief, or habit – not an external obstacle and thus not an excuse. For example, it is not the boss standing in the way, it is my resentment towards my boss which is the problem. Then I can act. If it is the boss themselves, I will feel helpless as I cannot change my boss. The internality of the obstacle needs to be stressed across the instructions.

It would be important, after participants have done the WOOP exercise, to stress that WOOP is an exercise that can be used for life-changing wishes but also in daily life to structure the upcoming day, the upcoming meeting, the long-postponed phone call, the upcoming evening. It is also important to stress that WOOP can and should be used every day, possibly at the same time, for all sorts of wishes and concerns. It is a skill that needs to be practiced. This guidance for the usage of WOOP should be included in the instructions.

It is unclear what the interactions between the interventionist and the participants are in the group context; it is also not described how the members of the group are supposed to communicate. Plus, is WOOP given in writing, or with spoken instructions, or a mixture? Again, the question is how standardized the instructions are, a question that is particularly relevant in case the instruction would vary between the different organizations. Randomization is, as I understand by workplace. If that is the case, will workplace vary with other variables? For example, interventionist, group size, intervention context, setting, etc.? If these confounds exist, it would be even more important to standardize instructional materials.

As for the dependent variables, why would the authors not try to record objective measures? It has been shown that objective measures are often a better indicator of behavior change after mental contrasting and WOOP (because when using WOOP automatic processes that people cannot report about are responsible for the behavior change) than subjective measures. Would the authors be able to ask other people observe the participants (within the group?), retrieve some performance (e.g., performance evaluations), and well-being indicators (e.g., sick days)?

In sum, the instructions around WOOP should be made more consistent and in line with the theory and the literature (as guidance see woopmylife.org). The terminology of goal setting may be misleading, and the examples should be adjusted and standardized. In that respect, examples should be provided sparsely. People tend to emulate them ending up with wishes and obstacles that they do not feel strongly about.

If questions about details arise, the authors may turn to woopmylife.org, and initiate contact there.

7. PLOS authors have the option to publish the peer review history of their article (what does this mean?). If published, this will include your full peer review and any attached files.

Reviewer #1: **Yes: **Jeremy Franklin

Reviewer #2: No

---

## [Author Response · Author response to Decision Letter 0]

7 Oct 2022

***Indicates author reply

Reviewer #1: This manuscript describes a proposed study design and protocol aiming to evaluate a mental contrasting intervention for achievement of health and well-being goals in the workplace.

In contrast to the stated rationale of showing an effect on health behaviour (p. 3-4), the objectives of the study (p.5) are all concerned with the participants’ perception of health and health-related behaviour. Indeed, the main outcomes are participants’ self-reported evaluation of their progress with respect to the goals they set at baseline. Several factors may be expected to influence these perceptions, including the expectations which the participants associate with the intervention, to which they are not blinded. Therefore, the results will not allow direct conclusions concerning the actual, objective effects on health and health behaviour, which will not be determined.

While these limitations apply to many studies especially of psychological interventions, the very subjective nature of the outcomes and the fact that participants are fully aware of the aims of the study and of their allocated intervention make them especially acute in this proposed study. They might be mitigated by veiling the study aim for the participants prior to the intervention, as seems to have been done by Oettingen, Mayer and Brinkmann (Ref. 18), and/or by asking the participants to assess their current health status and behaviour rather than related to baseline. The use of a concomitant ‘placebo’ intervention in the wait-list group might help to reduce disparity in expectations in the two groups. Failing this, these limitations should be described more fully and the planned procedure and questionnaire should be justified in detail.

***Thank you for raising the important points about the use of self-reported health rather than objective health, expectations participants associate with the intervention, and mitigation options. While we would like to use measures of objective health, we are limited by practical constraints (access to the workplaces) and resource constraints (funding availability). We have expanded where we discuss the limitations of self-reported health: “Finally, all outcomes are self-reported. This is a limitation because no conclusions can be drawn about objective effects on health or health behaviour, and there could be response bias.”

***While we would like to fully veil the study aim as by Oettingen, Mayer and Brinkmann (Ref. 18), our experience with employers was that they wanted to know the type of content within the sessions before committing to them. We do ask that employers not share the description of the sessions with staff, rather describing them more generally as ‘goal setting sessions’ or ‘going from ‘want’ to ‘do’’. In our questionnaire, we ask if participants have heard of WOOP or mental contrasting plus implementation intentions before (S8 – Q20). Thus, we will be able to assess if the study aim was sufficiently protected.

***Following the advice of Reviewer 2, our new approach to a concomitant ‘placebo’ intervention in the wait-list group is to focus on the ‘W’ of WOOP and go through its key components (see new S3). Thus we are comparing the ‘W’ of WOOP (received by both intervention and control in the first session) versus the ‘OOP’ (received only by the intervention group in the first session). Participants will not be told if they are in the intervention group, which is now clarified under the section Blinding (masking). 

It is important that the choices of self-set goal at baseline are broadly comparable between the intervention and waitlist groups. If possible, participants should remain blind to allocation until these goals have been set. At data analysis, choices should be presented descriptively (e.g. by building categories) and compared between groups.

***We agree this is important. The researcher will not be aware of the group allocation until the goals are set – thus, neither will the participants, and the participants are never told if they are in the intervention or control group. We now include descriptive choices for goals (see S8, Q25: What was your wish about? Health, social relationships, work performance, academic, other). 

Minor comments:

1. Please explain why a Bayesian (linear mixed) model is planned, since there does not seem to be a concretely relevant prior for the effect.

***Thank you for your comment. The use of Bayesian analysis using an uninformative priors allows us to provide posterior probabilities for the effect of the intervention instead of dichotomising the results based on p-values.

2. Multiple imputation (p.15) is proposed for dealing with missing covariates in a secondary analysis. It would be good to give a reference for this technique.

***Thank you for your comment. A reference for the multiple imputation methods has been added: Carpenter J, Kenward M. Multiple imputation and its application. John Wiley & Sons; 2012 Dec 21.

3. A statistician is supposed to check data for accuracy (p.16) – is this possible, or do you mean plausibility?

***Thank you for your comment. Yes you are correct and the text had been updated to reflect this.

Reviewer #2: Review of Registered Report Protocol PONE-D-22-11119 submitted to PLOS ONE entitled: “A cluster randomised wait-list controlled trial of a goal-setting behavior change intervention for employees in workplaces enrolled in workplace health and well-being initiatives”.

The present protocol describes an intervention study geared at supporting behavior change in the workplace context. The intervention employs a behavior change strategy called WOOP (Wish, Outcome, Obstacle, Plan) which in the scientific literature stands for mental contrasting (MC or WOO) by itself or combined with implementation intentions (MCII or WOOP). Mental contrasting with or without implementation intentions is well tested in the health domain but hardly in the work domain. In fact, the proposal argues that the innovation of the proposed study is that only three studies focus on mental contrasting or WOOP in the workplace and has never been systematically examined in a broad sample of workers during their everyday lives. Impressively, the authors want to address 60 organizations with 30 organizations in the intervention group and 30 organizations in the waitlisted control group. The wishes are supposed to tackle desired futures from the health and well-being domains.

The protocol includes a host of details that is relevant for administrative purposes, is well written, and gives an overview about what the authors are planning to do. The research question asking for the acceptance and effects of WOOP in the workplace is very important, and because the protocol aims at a relatively large sample of organizations and participants, the study is well worth to be conducted. The materials are in the most part well described.

Unfortunately, I am a bit unclear about my role as a reviewer. Judging from the timetables, the study has already started. The question then is, how much can the authors still change in the protocols? I think the study should be done, especially after all the effort of obtaining the contacts to the organizations. However, they are still a host of details which should be improved when it comes to design and instructions of the intervention. In the following, I provide examples on how the materials could be refined according to the theoretical and empirical principles of WOOP. I hope refinement is still possible before the beginning of the study.

***Thank you for the helpful suggestions to refine, which we were able to incorporate and pilot since receiving your comments. We detail our revisions below.

I might not fully understand in what order and in what time frame the instructions are given. Do the authors assume that they can have the baseline filled out and the WOOP exercise taught in 45 minutes to an hour? Have the authors piloted that time frame? I'm also unclear how many participants should be in one group, how the researchers are trained to convey WOOP, how often WOOP is practiced in the first instruction session, and how the materials (participants take them home?) are explained. Plus, what are the fidelity criteria, etc.

***In terms of timeframe, yes, we piloted the timeframe for the questionnaire and the WOOP exercise. Thus far we have not exceeded 50 minutes for both.

***In terms of group size, we expect that there will be 10-30 employees in each group: “The key contacts in organisations will be asked to invite volunteer employees to participate until there are at least 10 and no more than 30 employees per organisation.”

***In terms of training for the researcher to convey WOOP, we have updated the information on ‘Who’ in S4:

***Who provided. A member of the research team (MSc+) familiar with literature and use of mental contrasting, implementation intentions, and WOOP interventions will provide the sessions. They will have PowerPoint slides and some text to follow during the session. All members of the research team running the sessions will watch videos and listen to audios from the WOOP website (woopmylife.org), attend a training session run by [name of co-author, blind for peer review], and pilot the sessions – receiving feedback from other researchers. 

***WOOP is practiced once in the first instruction session and the handout materials are provided to participants so that they can take them home. Participants are told they can WOOP every day. To confirm that manipulation of OOP occurred as planned (fidelity), researchers delivering the sessions fill out a reporting form and one of the questions is, “Did you deliver the intervention or control session as indicated in the envelope?” This information is now updated in S4.

It would be important that the interventionists strictly adhere to the instructions and that the instructions are better standardized. WOOP is a sensitive animal and small changes in wording might have big effects, as the experimental data show. In that respect, as nice as it is that the authors want to give freedom in delivery to the interventionists and want to leave the setting unspecified (Zoom, in person), there is the danger of diluting the WOOP effects. Experimental control should be guaranteed, at least as much as possible. Certainly, we need a balance between the autonomy of the interventionist and the sensitivity of the WOOP exercise, but it would be helpful to twist the WOOP instructions and examples (especially at the beginning) more in the direction of being in line with the WOOP principles plus towards being standardized. To find a compromise between standardization and autonomy of the interventionist the authors may consult the woopmylife.org website. Here one can also speak to the researchers who generated the strategy.

***Thank you for the suggestion to better standardise the instructions. Previously, we had followed National Institute for Health Research principles and guidance on adapting interventions for a local context using the advice of public contributors (https://www.nihr.ac.uk/health-and-care-professionals/engagement-and-participation-in-research/involve-patients.htm). However, we appreciate that a balance between the autonomy of the interventionist and the sensitivity of the WOOP exercise are needed.

***As suggested, we contacted the woopmylife.org website to ask about standardization versus the autonomy of the interventionist. We gratefully received the advice that: “it is crucial to watch out for the standardization of the features WOOP that are responsible for the effects (e.g., four steps, order of the four steps, imagery, specificity, etc.).” In our intervention, the four steps, order of the four steps, use of imagery, and specificity were preserved. 

***After careful deliberation, including discussion with our public involvement liaison and co-authors, we decided to adhere to the material more close to prior WOOP studies that align with material on on woopmylife.org. We could not use the exact website material is it is copyrighted. The studies we used to inform the intervention materials were:

***Schweiger Gallo I, Bieleke M, Alonso MA, Gollwitzer PM, Oettingen G. Downregulation of anger by mental contrasting with implementation intentions (MCII). Front Psychol. 2018 Oct 4;9:1838.

***Von Weichs V, Krott NR, Oettingen G. The Self-Regulation of Conformity: Mental Contrasting With Implementation Intentions (MCII). Front Psychol. 2021 Jun 2;12:546178.

***Gollwitzer PM, Mayer D, Frick C, Oettingen G. Promoting the self-regulation of stress in health care providers: An internet-based intervention. Front Psychol. 2018 Jun 15;9:838.

***We revised the Powerpoint slides and notes and re-piloted the sessions. One of our considerations was our desire to conduct an independent evaluation to guide scientific results and general conclusions (Lilford, 2018). Independent evaluations add credibility by having those who designed the intervention separate from those who evaluated it – reducing potential bias. The new sessions are in S3 and some of the changes are described below in response to the suggested changes to the instructions. 

***Unfortunately, it was not possible to standardise the mode of delivery. Some workplaces on the local government programme require virtual sessions because their employees are remote, whereas others required in-person sessions. We find the workplaces are unwilling to participate if they cannot choose. 

***Lilford RJ. Implementation science at the crossroads. BMJ quality & safety. 2018 Apr 1;27(4):331-2.

Let me specify some points to change in the instructions (without that I can be exhaustive here):

First the authors talk about a goal setting intervention. WOOP, however, starts with the Wish, not with the goal, and the research on mental contrasting makes a strong point that people generate wishes and concerns about desired futures rather than goals. In fact, only after mental contrasting wishes might turn into committal goals. It is critical that participants understand that they come up with a wish, that they are searching for. For finding their wish, participants need to be calm and slow, need to understand that their wish should be “dear to their heart” and truly important. By simply naming a goal participants will not achieve finding this wish (as we have so many goals on the top of our heads). The first step in WOOP, then, is foraging for a wish. Please consult the woopmylife.org website for relevant instructions. The authors can download the WOOP kit and for the tone and slowness also watch the instructional videos. In the instructions of the present protocol, the introduction of the goal at the beginning and then making the first WOOP regarding that goal might mislead people.

In the same vein, it would be helpful if the authors relinquish the term goal setting. The term goal often implies being overladen with pressure and demands from the outside world. Therefore, if the study still can be changed, it would be helpful to let go from the term goal setting when introducing the instructions. Also, to not confuse people, the term WOOP instead of MCII might be enough. Two terms are an unnecessary load.

***Thank you for these suggestions.

***Regarding finding the wish, we agree that we need to ensure more slowness and mimic the tone on woopmylife.org. Thus, all researchers delivering the sessions now watch the videos and listen to the audios on woopmylife.org. We have taken the wish out of the survey in order to slowly guide participants in the session through finding something truly important and dear to them. The main part of the first session that all participants receive is now thinking about and reflecting on their wish, including its key components (Truly important? Possible to achieve? Challenging? Summarised in no more than a few sentences?) We now ensure plenty of time to really find something that is important to participants (see S3a).

***Regarding the word goal, we initially included ‘goal’ rather than ‘wish’ on the advice of our Patient and Public Involvement contributors. The contributors had stated that people in the sessions might find the word ‘wish’ to be childish. However, we agree that the word goal could have implications related to pressure and demands from the outside world. Thus, we have largely reverted to using the word ‘wish’.

***On the point about WOOP instead of MCII, we now only use WOOP to reduce cognitive load. 

When introducing WOOP, the examples do not comply with the WOOP principles. They should be referring to wish, outcome, obstacle, and behavior to overcome the obstacle. These four steps should be built on each other. And they should be short and concise (later the instructions talk about 3 to 6 words). Again, please check the woomylife.org website.

***We have removed all examples to avoid biasing participants. Again, our Patient and Public Involvement contributors thought that having real examples could enhance the credibility of the intervention. Upon reflection, the desire to reduce bias outweighed the desire to enhance the credibility of the intervention.

Later the instructions are more in line with the WOOP principles (p. 56 left side). The authors may want to adjust the examples and instructions, make them consistent across the study, and consistent with what has been depicted on the website.

***The instructions have been revised to align with previously published WOOP material and be consistent across the study. All examples have been removed. To avoid biasing in line with these examples. 

But even in the detailed WOOP instructions, the authors might check again: For example, on p. 51 in the instructions the step to find and specify the behavior to overcome the obstacle is missing. This is a crucial step and should be integrated. Or, in the booklet, it is unclear what people need to do, how often, when, etc. Why does the wish have so many lines? P. 56, too many if-then plans might hurt.

***The booklet has been removed. Instead, participants are given Wish and OOP handouts. The step to find and specify the behaviour to overcome the obstacle is now included. 

The preparation for the WOOP exercise is missing. To do a WOOP exercise, people need to be slow and calm, and the instructions before the exercise need to explicitly say that. Otherwise, participants will not generate a meaningful wish and will not be able to come up with the best outcome, and internal critical obstacle, nor will they find a behavior to overcome the obstacle. Importantly, they will not be able to vividly imagine the outcome and the obstacle either.

***Thank you for pointing out this important issue. We agree and now create space for participants to be slow and calm before their wish, see S3a, p5. 

It is crucial that people understand that the obstacle needs to be an inner obstacle, an obstacle such as an emotion, belief, or habit – not an external obstacle and thus not an excuse. For example, it is not the boss standing in the way, it is my resentment towards my boss which is the problem. Then I can act. If it is the boss themselves, I will feel helpless as I cannot change my boss. The internality of the obstacle needs to be stressed across the instructions.

***We agree and now use language around inner obstacle such as emotions, beliefs, or habits, see S3b, pages 6-7, S3e, pages 7-9.

It would be important, after participants have done the WOOP exercise, to stress that WOOP is an exercise that can be used for life-changing wishes but also in daily life to structure the upcoming day, the upcoming meeting, the long-postponed phone call, the upcoming evening. It is also important to stress that WOOP can and should be used every day, possibly at the same time, for all sorts of wishes and concerns. It is a skill that needs to be practiced. This guidance for the usage of WOOP should be included in the instructions.

***We agree and now use similar language without infringing on the woopmylife.org copyright (we did ask to use the material more precisely but received no reply):

***“You can use WOOP as much or as little ask you like – for small wishes and big wishes, short term wishes, or long term wishes. You could use WOOP when you feel like you have too much on your plate or when there is something you would really like to do for yourself. Some people say they use WOOP everyday and that it helps them in their everyday life and plans for their future. Do not worry if every wish you have does not happen, though. People who use WOOP say they have seen improvements in their work, family life, and relationships. If you use WOOP often, and practice it, you will get better at it. So really get to know it - it can help you to find out what it is you really want and what is it that keep you from going for what you really want. You can go from 'wanting' - and knowing more about your 'wanting' - to doing.” 

It is unclear what the interactions between the interventionist and the participants are in the group context; it is also not described how the members of the group are supposed to communicate. Plus, is WOOP given in writing, or with spoken instructions, or a mixture? Again, the question is how standardized the instructions are, a question that is particularly relevant in case the instruction would vary between the different organizations. Randomization is, as I understand by workplace. If that is the case, will workplace vary with other variables? For example, interventionist, group size, intervention context, setting, etc.? If these confounds exist, it would be even more important to standardize instructional materials.

***We now have clarified beneath the slides where interaction points occur. These were re-developed after re-piloting. In the S3 material, every point of interaction is now noted as 

"**’Interaction’"

***To summarise, there are several main points for interaction between interventionist and participants: 

• In the beginning, they are asked about common intention-behaviour gaps: how many of you have wanted to do something to improve your health and wellbeing but it has not happened in the end? (show of hands) S3a, page 2. 

• Discussion of the usefulness of WOOP (see S3b, page 13; see S3c, pages 1-3).

• In the second sessions, the questions from normalisation process theory are interactive (see S3c, page 2; S3d, pages 2-6).

***WOOP is provided in writing (W and OOP handouts) and participants are taken through these handouts with spoken instructions. Instructions are the same between organisations, apart from the survey and handouts being delivered virtually in the virtual sessions versus on paper in the in-person sessions. 

***Yes randomisation is by workplace – random block sizes of two and four will be used to maintain balance.

As for the dependent variables, why would the authors not try to record objective measures? It has been shown that objective measures are often a better indicator of behavior change after mental contrasting and WOOP (because when using WOOP automatic processes that people cannot report about are responsible for the behavior change) than subjective measures. Would the authors be able to ask other people observe the participants (within the group?), retrieve some performance (e.g., performance evaluations), and well-being indicators (e.g., sick days)?

***In our previous work, we asked employers about the feasibility of collecting objective data and found it would be difficult within our time and funding constraints. Organisations in the local government programme tend to keep different types of records about sickness absence and performance – thus, the heterogeneity would make it difficult data to analyse. Employees reporting on each other may raise privacy concerns. We note that “future research could examine the effects on observational measures like absenteeism or productivity”, although this could require organisations standardising their approach in advance. We would like to explore this in our future work.

In sum, the instructions around WOOP should be made more consistent and in line with the theory and the literature (as guidance see woopmylife.org). The terminology of goal setting may be misleading, and the examples should be adjusted and standardized. In that respect, examples should be provided sparsely. People tend to emulate them ending up with wishes and obstacles that they do not feel strongly about.

If questions about details arise, the authors may turn to woopmylife.org, and initiate contact there.

***Thank you for your detailed and attentive comments and suggestions. The term goal has been largely removed and the examples are no longer included to avoid bias. We contacted woopmylife.org and included their suggestions. We believe our sessions are much stronger as a result.

---

## [Decision Letter · Decision Letter 1]

1 Dec 2022

PONE-D-22-11119R1A cluster randomised waitlist-controlled trial of a goal-setting behaviour change intervention for employees in workplaces enrolled in workplace health and wellbeing initiativesPLOS ONE

Dear Dr. Kudrna,

Thank you for submitting your manuscript to PLOS ONE. After careful consideration, we feel that it has merit but does not fully meet PLOS ONE’s publication criteria as it currently stands. Therefore, we invite you to submit a revised version of the manuscript that addresses the points raised during the review process.

Unfortunately, neither of the original reviewers were available to review your revised submitted. The manuscript has been evaluated by two new reviewers, and their comments are available below.

Both reviewers have made a number of requests for clarifications. 

Could you please carefully revise the manuscript to address all comments raised?

We look forward to receiving your revised manuscript.

Kind regards,

Steve Zimmerman, PhD

Associate Editor, PLOS ONE

Journal Requirements:

Reviewers' comments:

Reviewer's Responses to Questions

**Comments to the Author**

1. Does the manuscript provide a valid rationale for the proposed study, with clearly identified and justified research questions?

Reviewer #3: Yes

Reviewer #4: Partly

2. Is the protocol technically sound and planned in a manner that will lead to a meaningful outcome and allow testing the stated hypotheses?

Reviewer #3: Partly

Reviewer #4: Yes

3. Is the methodology feasible and described in sufficient detail to allow the work to be replicable?

Reviewer #3: No

Reviewer #4: Yes

4. Have the authors described where all data underlying the findings will be made available when the study is complete?

Reviewer #3: Yes

Reviewer #4: Yes

5. Is the manuscript presented in an intelligible fashion and written in standard English?

Reviewer #3: Yes

Reviewer #4: Yes

6. Review Comments to the Author

You may also provide optional suggestions and comments to authors that they might find helpful in planning their study.

Reviewer #3: The author has incorporated much of the responses of the previous reviewers which has clarified and addressed some of the concerns and comments. However there are still a few places where it could benefit from further clarification, such as:

>>The timing of the intervention/control sessions - will they be in worktime or not? It is important in a workplace study to be clear about when and where an intervention is being held - this needs to be standardised across the study

>>There is mention of a midline as a data collection timepoint (under statistician will check the data) but there is not a description or reference to this in the study description earlier - it seemed to be only 2 data collection points.

>>It is not clear what the participants will be doing in the 4+ weeks between start and finish - will they be carrying out the woop strategy whenever they wish? I think the study would benefit from some sort of quantifying of participant use of the intervention over the study period.

>>There is no recognition of the possibility of the participants being involved in other workplace interventions - this is a strong possibility especially as the workplaces being chosen are all those which are signed up for WHIs. It should be an exclusion criteria

>> In the first session the first action would be better and less prone to bias if the surveys were carried out prior to any discussion (as they are in the follow up session)

>> I wonder if the demographic questions are using standardised questionnaires - asking yes/no to have you a disability seems that it may not yield very helpful information.

>>How confident are you in the participants remembering their goals (can you put in place a system to remind them?) and does it matter if they change their goal or achieve one and go onto another one? Can you capture that?

>> In light of the difference between wish and goal, the title, and elsewhere in the text, would be better reworded to reflect this with something like 'wish derived goals'. It would be interesting to unpack this a bit more too.

>> The limitations of participant selection is partly acknowledged but there is also purposeful selection and the bias effect of this should be explained

>> Please clarify what 'all organisation involved in the research will receive extra help implementing content about health and wellbeing at work' - is this as an incentive to participate? Will it impact on the study at all?

Reviewer #4: Review of Registered Report Protocol PONE-D-22-11119 submitted to PLOS ONE entitled: “A cluster randomised wait-list controlled trial of a goal-setting behavior change intervention for employees in workplaces enrolled in workplace health and well-being initiatives”.

Thank you for your invitation to review this study protocol.

I acknowledge and appreciate the authors’ willingness to adapt their methods in response to the feedback from Reviewers 1 and 2. Overall, this protocol is clear and well written. A strength of this study is the large group of workplaces included in the sample, and the use of a placebo.

The authors do not describe how “mental contrasting” and “implementation intentions” differ from “goal setting” and “strategies” which are commonly used and widely evaluated health promotion concepts, including in workplace settings. For example, Malik, S. H., Blake, H., & Suggs, L. S. (2014). A systematic review of workplace health promotion interventions for increasing physical activity. British Journal of Health Psychology, 19(1), 149–180. https://doi.org/10.1111/bjhp.12052. How does this study differ from these previous studies? As Reviewer 2 stated, mental contrasting focuses on a “wish” as opposed to a goal, yet the term goal is still used throughout this protocol, including in the title. A wish and a goal are distinctly different constructs and therefore using them interchangeably creates confusion for the reader, and diminishes the novelty of the study.

I agree with reviewer one that I have concerns about the authors’ capacity to evaluate effects on health behaviour using the methods described. You state in the abstract that “… rigorous effectiveness evaluations of them are rare. In this research we propose evaluating the effectiveness of an established behaviour change intervention in a new workplace context.” It appears that the study will not evaluate clinical effectiveness or objective measures of behaviour change, but only the participants’ perception of effectiveness over a short timeframe, and therefore I find this language to be an overstatement as it gives the impression that this is a rigorous effectiveness evaluation. If inclusion of objective outcome measures, such as examples or observations of behaviour change, is not possible, I encourage the authors to be more circumspect in their language.

The “post” data collection is only four weeks after the “intervention” – is there any intention to track behaviour change over a longer period? Sustaining a behaviour change is more difficult than creating a short-term change over a four-week period. Is there evidence that a behaviour change over a four-week period is clinically significant for a person’s health outcomes without subsequent follow-up?

Thank you for including sex and gender as separate variables.

P14-15 - I will leave it to other reviewers more familiar with the Bayesian approach described to comment on these methods.

P 13 and 15 - Qualitative methods – there is a lack of detail about the qualitative analysis methods for Please include more details about your qualitative analysis methods such as how many coders, how will you ensure inter-rater reliability, will transcripts be deidentified prior to analysis, how will you address any potential bias of the coders and their interpretation of the data? For example, are the coders the same people conducting the workshops? Are they independent researchers? Please refer to a relevant qualitative research reporting standard e.g., O’Brien, Bridget C. PhD; Harris, Ilene B. PhD; Beckman, Thomas J. MD; Reed, Darcy A. MD, MPH; Cook, David A. MD, MHPE. Standards for Reporting Qualitative Research: A Synthesis of Recommendations. Academic Medicine: September 2014 - Volume 89 - Issue 9 - p 1245-1251 doi: 10.1097/ACM.0000000000000388

Fidelity – you have responded to Reviewer 2 that you will now ask the workshop facilitators whether they delivered the intervention or control session as indicated in the envelope. Will you also ask about any adaptations made and the reasons for these, so that these are documented? Although facilitators can have the best possible intentions, adaptations do happen and should be recorded.

Overall, I feel that the methods described have strengths, such as the large sample size, clear procedures which increase the likelihood of fidelity, and the variety of workplaces. However, there are also significant limitations related to the short timeframe between intervention and follow-up, and the lack of objective measures of behaviour change or clinical outcomes.

7. PLOS authors have the option to publish the peer review history of their article (what does this mean?). If published, this will include your full peer review and any attached files.

Reviewer #3: No

Reviewer #4: No

---

## [Author Response · Author response to Decision Letter 1]

17 Jan 2023

17 January 2023

Dear Editor,

Thank you and the reviewers for reading our manuscript and providing suggestions to improve it. Please note that data collection for the trial has now commenced. Thus, we re-submit our manuscript as a Protocol rather than a Registered Report, following advice from the Editorial Desk. Prior to data collection, we registered our trial on a trial registry (ISRCTN17828539) and deposited the previously submitted version of this protocol on the Open Science Foundation (version 2.0 19.09.22). 

Our reply to the reviewers is below. The main changes include clarifying the timing within the sessions and aspects of our data collection tools, further detailing our approach to qualitative analysis, and collecting data on any adaptations to the planned intervention delivery. We now note several limitations in more detail, including those related to bias from self-report measures, the short follow-up period, and purposeful participant selection. Following the reviewers’ advice, we considered the differences between wishes and goals in more depth and now provide text about their relationship. Accordingly, the text of our title has changed from a ‘goal-setting’ to a ‘goal-based’ intervention.

We believe we have addressed the issues raised by the reviewers and look forward to hearing from you.

Sincerely,

Dr Laura Kudrna (on behalf of co-authors)

>>Reviewer #3: The author has incorporated much of the responses of the previous reviewers which has clarified and addressed some of the concerns and comments. However there are still a few places where it could benefit from further clarification, such as:

>>The timing of the intervention/control sessions - will they be in worktime or not? It is important in a workplace study to be clear about when and where an intervention is being held - this needs to be standardised across the study

We agree and have clarified that the intervention will be “delivered to employees from the same organisation in a group setting during working hours.”

>>There is mention of a midline as a data collection timepoint (under statistician will check the data) but there is not a description or reference to this in the study description earlier - it seemed to be only 2 data collection points.

Thank you for noticing this typo, we have removed reference to the midline. 

>>It is not clear what the participants will be doing in the 4+ weeks between start and finish - will they be carrying out the woop strategy whenever they wish? I think the study would benefit from some sort of quantifying of participant use of the intervention over the study period.

This issue was also raised during our piloting and participants in the intervention group are told they can keep using WOOP (see S3b, page 12):

“You can use WOOP as much or as little ask you like – for small wishes and big wishes, short term wishes, or long term wishes. You could use WOOP when you feel like you have too much on your plate or when there is something you would really like to do for yourself. Some people say they use WOOP everyday and that it helps them in their everyday life and plans for their future. Do not worry if every wish you have does not happen, though. People who use WOOP say they have seen improvements in their work, family life, and relationships. If you use WOOP often, and practice it, you will get better at it. So really get to know it - it can help you to find out what it is you really want and what is it that keep you from going for what you really want. You can go from 'wanting' - and knowing more about your 'wanting' - to doing.”

Although we do not ask people how many times they used WOOP, we do ask participants in the group discussion if their WOOP was successful and why, which does raise the issue of how much they are carrying it out (see S3d, page 5):

“Now, think about your wish. In your opinion, did you attain it? If so, how? If not, why?” 

In these discussions, participants are mentioning if they are using WOOP although we do not ask how often specifically. Unfortunately, we have now started data collection and cannot add measures without deviating from our planned approach. Instead, we note the issue of the intervention ‘dose’ as a limitation: 

“The dose of the intervention in one group session might be too weak and repeated practice or reminders over the study period could strengthen its effects.”

>>There is no recognition of the possibility of the participants being involved in other workplace interventions - this is a strong possibility especially as the workplaces being chosen are all those which are signed up for WHIs. It should be an exclusion criteria

Thank you for raising this important point. We agree it would be ideal to achieve full experimental control by excluding workplaces offering other interventions. However, all the workplaces in the study will be offering other interventions because they are participating in the local government health and wellbeing scheme. Our funding requires us to partner with the local government scheme to achieve policy impact.

The WOOP intervention sits alongside a menu of other initiatives in the scheme, such as walking clubs, anti-bullying policies, and free fruit and vegetables in kitchens. We unfortunately cannot exclude workplaces participating in the local government schemes because there would be no workplaces to participate in the study. While this is a weakness from an experimental design point of view, it does enhance the ecological validity of the study because it sits alongside ongoing initiatives. We now note this as a limitation:

“There could also be interactions between the intervention and other ongoing initiatives within the workplaces.”

>> In the first session the first action would be better and less prone to bias if the surveys were carried out prior to any discussion (as they are in the follow up session)

We carry out all surveys prior to any discussion (see Table 1 – a and b), which is now clarified in the text: “Note that all surveys are carried out prior to discussion in both sessions, meaning endpoint measures will be taken from the control group before they receive their intervention.”

>> I wonder if the demographic questions are using standardised questionnaires - asking yes/no to have you a disability seems that it may not yield very helpful information.

We updated the manuscript to reflect that if participants say they have a disability they are asked to expand. The demographic questions are drawn from the national census survey. 

>>How confident are you in the participants remembering their goals (can you put in place a system to remind them?) and does it matter if they change their goal or achieve one and go onto another one? Can you capture that?

We are not sure if everyone will remember, and we have accounted for this in our questionnaire text:

“As a reminder, in the previous session, you may have been asked to tell us about something that you wished to do for your health and wellbeing. Do you remember being asked about this? Yes/No/Unsure. As a reminder: It was supposed to be something challenging, but feasible to achieve in around four weeks. It did not need to be work-related, but it should have been something that you thought that you could complete in around four weeks. What did you wish to do for your health and wellbeing? (if you do not remember, leave this blank).”

We agree it would be interesting to capture if participants change their goal or achieve one and go onto another one. However, the instructions ask participants to focus on their most important wish if they have several. Therefore, we focus on their most important wish. We do ask about their cognitive participation with WOOP (incorporate it into work life and life outside work). The focus on single wish is a limitation we now acknowledge: “Multiple doses for various wishes could be required to achieve results.”

The issue of reminders was also raised in our piloting and, as a result of these discussions, participants now take away a WOOP handout where they write down their WOOP (see S3f). Unfortunately, we did not have the resources to implement a more sophisticated reminder system with our budget. Another consideration was that not all employees were easily contactable for reminders - some of them do not have work email addresses (they might work as drivers or in factories). We decided not to ask them for their personal phone numbers or email addresses for privacy reasons. Thus, we decided to leave staff with the handouts and leave it up to them to use them.

>> In light of the difference between wish and goal, the title, and elsewhere in the text, would be better reworded to reflect this with something like 'wish derived goals'. It would be interesting to unpack this a bit more too.

Thank you for this comment. We agree it is important to further specify the link between wishes and goals to clarify our approach. In the existing literature on mental contrasting plus implementation intentions, wishes are potential goals, and wishes become goals once they are selected and people commit to them. Thus, upon reflection, we perceive this is a goal-based intervention because participants are asked to commit to their wishes, turn them into goals, and report on whether they were able to achieve them. Our title has been changed accordingly to goal-based rather than goal-setting and we have clarified our use of the language within the manuscript text.

To clarify our use of this language, the introduction, we now link wishes to goals by stating: “In applications of mental contrasting and implementation intentions, participants are asked to select from multiple potential goals (called wishes) and strengthen their commitment to them using these strategies.” We also now describe wishes and goals in the intervention section: “During the intervention activity, participants will be encouraged to use mental contrasting and implementation intention steps to reinforce the wish, turning it into a committed goal.”

We draw on text from Gollwitzer, P. & Oettingen, G. (2011):

“The concept of implementation intentions grew out of a more comprehensive approach to goal pursuit: the mindset theory of action phases (Gollwitzer, 1990). The mindset model of action phases sees successful goal pursuit as solving a series of successive tasks: deliberating on wishes (potential goals) and choosing between them; planning and initiating goal-directed actions; bringing goal pursuit to a successful end; and evaluating its outcome. This task notion implies that people can activate cognitive procedures (mindsets) that facilitate task completion simply by getting heavily involved with the task at hand. Whereas deliberating between potential goals (i.e., wishes) activates cognitive procedures (i.e., a deliberative mindset) that facilitate decision making, engaging in planning activates those procedures (i.e., an implemental mindset) that support the implementation of goals (p. 177).”

Gollwitzer, P. & Oettingen, G. (2011). Planning promotes goal striving. In K. Vohs & R. Baumeister (Eds.), Handbook of self-regulation: research, theory, and applications (pp. 162-185). ISBN 978-1-606-23948-3

>> The limitations of participant selection is partly acknowledged but there is also purposeful selection and the bias effect of this should be explained

We have modified the text to explain the effect of purposeful selection on generalisability:

“A further limitation is that employees volunteered to take part, which may introduce selection bias. Although a mitigation step is planned by asking workplaces to invite staff across diverse occupations in workplaces, purposeful selection in this way means that the results may not generalise to those who are not selected.”

>> Please clarify what 'all organisation involved in the research will receive extra help implementing content about health and wellbeing at work' - is this as an incentive to participate? Will it impact on the study at all?

The extra help is the intervention itself. We do not specify what the intervention is prior to delivering it to avoid participants going through WOOP before the session with us. However, our ethics team wanted us to clarify that something extra will take place with the research.

>>Reviewer #4: Review of Registered Report Protocol PONE-D-22-11119 submitted to PLOS ONE entitled: “A cluster randomised wait-list controlled trial of a goal-setting behavior change intervention for employees in workplaces enrolled in workplace health and well-being initiatives”.

>>Thank you for your invitation to review this study protocol.

>>I acknowledge and appreciate the authors’ willingness to adapt their methods in response to the feedback from Reviewers 1 and 2. Overall, this protocol is clear and well written. A strength of this study is the large group of workplaces included in the sample, and the use of a placebo.

>>The authors do not describe how “mental contrasting” and “implementation intentions” differ from “goal setting” and “strategies” which are commonly used and widely evaluated health promotion concepts, including in workplace settings. For example, Malik, S. H., Blake, H., & Suggs, L. S. (2014). A systematic review of workplace health promotion interventions for increasing physical activity. British Journal of Health Psychology, 19(1), 149–180. https://doi.org/10.1111/bjhp.12052. How does this study differ from these previous studies? As Reviewer 2 stated, mental contrasting focuses on a “wish” as opposed to a goal, yet the term goal is still used throughout this protocol, including in the title. A wish and a goal are distinctly different constructs and therefore using them interchangeably creates confusion for the reader, and diminishes the novelty of the study.

Thank you for this important comment, which also aligned with Reviewer 3’s comment on wish-derived goals (see above). We have changed the title and manuscript text to better address the relationship between wish and goal. We now also provide further detail on how mental contrasting and implementation intentions differ from goal setting and strategies as health promotion concepts.

To recap, our title has been changed from goal-setting to goal-based because mental contrasting and implementation intentions are more than about setting goals. Instead, these strategies also strengthen goal commitment and action planning, which are additional to the process of goal setting. 

Drawing on Gollwitzer & Oettingen (2011), we now clarify the relationship between wishes and goals with the following text (see also above): “In applications of mental contrasting and implementation intentions, participants are asked to select from multiple potential goals (called wishes) and strengthen their commitment to them using these strategies.”

>>I agree with reviewer one that I have concerns about the authors’ capacity to evaluate effects on health behaviour using the methods described. You state in the abstract that “… rigorous effectiveness evaluations of them are rare. In this research we propose evaluating the effectiveness of an established behaviour change intervention in a new workplace context.” It appears that the study will not evaluate clinical effectiveness or objective measures of behaviour change, but only the participants’ perception of effectiveness over a short timeframe, and therefore I find this language to be an overstatement as it gives the impression that this is a rigorous effectiveness evaluation. If inclusion of objective outcome measures, such as examples or observations of behaviour change, is not possible, I encourage the authors to be more circumspect in their language.

We agree that we implied our evaluation was a rigorous effectiveness evaluation which is inconsistent with our reliance on self-report measures. We have removed the word rigorous from the Abstract and now pre-empt the reader in the Introduction to expect self-reported measures. However, we have kept the language of effectiveness evaluation because we include a control group and ask participants if they changed their behaviour. 

The text changes to the Introduction now reads: “Our outcomes are self-reported and the intervention period lasts only several weeks, however, if positive effects are found, a more rigorous trial could consider including objective measures of health and extending the timeframe.”

>>The “post” data collection is only four weeks after the “intervention” – is there any intention to track behaviour change over a longer period? Sustaining a behaviour change is more difficult than creating a short-term change over a four-week period. Is there evidence that a behaviour change over a four-week period is clinically significant for a person’s health outcomes without subsequent follow-up?

Although we would like to be able to track participants beyond four weeks, we do not have the resources for a longer follow-up period due to funding limitations. We agree that sustained behaviour change is a challenge to achieve and there would be questions about the clinical significance of any changes, especially given the heterogeneity of the goals participants can set for themselves. Our limitations section now expands on this point: “If positive effects are found for measures of self-report, future research could examine the effects on observational measures like absenteeism or productivity, if any changes are sustained over the longer term, or if changes have clinically significant implications for health and wellbeing.” 

>>Thank you for including sex and gender as separate variables.

>>P14-15 - I will leave it to other reviewers more familiar with the Bayesian approach described to comment on these methods.

>>P 13 and 15 - Qualitative methods – there is a lack of detail about the qualitative analysis methods for Please include more details about your qualitative analysis methods such as how many coders, how will you ensure inter-rater reliability, will transcripts be deidentified prior to analysis, how will you address any potential bias of the coders and their interpretation of the data? For example, are the coders the same people conducting the workshops? Are they independent researchers? Please refer to a relevant qualitative research reporting standard e.g., O’Brien, Bridget C. PhD; Harris, Ilene B. PhD; Beckman, Thomas J. MD; Reed, Darcy A. MD, MPH; Cook, David A. MD, MHPE. Standards for Reporting Qualitative Research: A Synthesis of Recommendations. Academic Medicine: September 2014 - Volume 89 - Issue 9 - p 1245-1251 doi: 10.1097/ACM.0000000000000388

Thank you for the opportunity to expand in detail about the qualitative analysis. As we are taking a reflexive thematic approach to identify patterns of shared meaning, we will not be evaluating inter-rater reliability (which is aligned with thematic analysis approaches using cluster and call coding reliability or thematic analysis with codebooks – see references 63-65). We will follow COREQ reporting standards and now specify this in our porotocol. Our updated text now reads:

“Analysis of qualitative data

Qualitatively, focus groups will be audio recorded, transcribed, and coded, and reflexive thematic analysis will be used to conceptualise analytic themes about barriers, enablers, and mechanisms surrounding the use of WOOP [63–65]. We also analyse fieldnotes from focus groups. To inform the development of themes, will draw on the behaviour change wheel[24] to deductively shape our understanding of the data on barriers and enablers and on normalisation process theory to shape themes from the data on mechanisms.[51] Our coding approach will involve members of the research team who both did and did not collect the data, mixing informed and independent approaches, taking a collaborative and reflexive approach to identify patterns of shared meaning across the dataset [65]. Our results will be reported using the consolidated criteria for reporting qualitative research [66].”

>>Fidelity – you have responded to Reviewer 2 that you will now ask the workshop facilitators whether they delivered the intervention or control session as indicated in the envelope. Will you also ask about any adaptations made and the reasons for these, so that these are documented? Although facilitators can have the best possible intentions, adaptations do happen and should be recorded.

Thank you for this good idea and we have now updated our reporting from to ask about any adaptations made and the reasons for these. For sessions already conducted, session facilitators have been asked to update their forms. 

>>Overall, I feel that the methods described have strengths, such as the large sample size, clear procedures which increase the likelihood of fidelity, and the variety of workplaces. However, there are also significant limitations related to the short timeframe between intervention and follow-up, and the lack of objective measures of behaviour change or clinical outcomes.

Thank you for your suggestions to improve our study.

---

## [Decision Letter · Decision Letter 2]

24 Feb 2023

A cluster randomised waitlist-controlled trial of a goal-based behaviour change intervention for employees in workplaces enrolled in health and wellbeing initiatives

PONE-D-22-11119R2

Dear Dr. Kudrna,

We’re pleased to inform you that your manuscript has been judged scientifically suitable for publication and will be formally accepted for publication once it meets all outstanding technical requirements.

Kind regards,

Vidanka Vasilevski

Academic Editor

PLOS ONE

Reviewers' comments:

Reviewer's Responses to Questions

**Comments to the Author**

1. Does the manuscript provide a valid rationale for the proposed study, with clearly identified and justified research questions?

Reviewer #4: Yes

2. Is the protocol technically sound and planned in a manner that will lead to a meaningful outcome and allow testing the stated hypotheses?

Reviewer #4: Yes

3. Is the methodology feasible and described in sufficient detail to allow the work to be replicable?

Reviewer #4: Yes

4. Have the authors described where all data underlying the findings will be made available when the study is complete?

Reviewer #4: Yes

5. Is the manuscript presented in an intelligible fashion and written in standard English?

Reviewer #4: Yes

6. Review Comments to the Author

You may also provide optional suggestions and comments to authors that they might find helpful in planning their study.

Reviewer #4: The authors have responded to my, and the other reviewers' comments, to my satisfaction. I wish them luck with this study and look forward to seeing the results.

7. PLOS authors have the option to publish the peer review history of their article (what does this mean?). If published, this will include your full peer review and any attached files.

Reviewer #4: No

---

## [Editor Report · Acceptance letter]

15 Mar 2023

PONE-D-22-11119R2 

Protocol for a cluster randomised waitlist-controlled trial of a goal-based behaviour change intervention for employees in workplaces enrolled in health and wellbeing initiatives 

Dear Dr. Kudrna:

I'm pleased to inform you that your manuscript has been deemed suitable for publication in PLOS ONE. Congratulations! Your manuscript is now with our production department. 

Kind regards, 

on behalf of

Dr. Vidanka Vasilevski 

Academic Editor

PLOS ONE